# Fair Evaluation of Graph Markov Neural Networks

## Abstract

Graph Markov Neural Networks (GMNN) have recently been proposed to improve regular graph neural networks (GNN) by including label dependencies into the semi-supervised node classification task. GMNNs do this in a theoretically principled way and use three kinds of information to predict labels. Just like ordinary GNNs, they use the node features and the graph structure but they moreover leverage information from the labels of neighboring nodes to improve the accuracy of their predictions. In this paper, we introduce a new dataset named *WikiVitals* which contains a graph of 48k mutually referred Wikipedia articles classified into 32 categories and connected by 2.3M edges. Our aim is to rigorously evaluate the contributions of three distinct sources of information to the prediction accuracy of GMNN for this dataset: the content of the articles, their connections with each other and the correlations among their labels. For this purpose we adapt a method which was recently proposed for performing fair comparisons of GNN performance using an appropriate randomization over partitions and a clear separation of model selection and model assessment.

## 1 Introduction

Graph neural networks (GNN) (Yang et al., 2016; Kipf & Welling, 2017; Defferrard et al., 2016) have become a tool of choice when modeling datasets whose observations are not i.i.d. but are comprised of entities interconnected according to a graph of relations. They can be used either for graph classification, like molecule classification (Dobson & Doig, 2003; Borgwardt et al., 2005), or for node classification, like document classification in a citation network (Sen et al., 2008).

The most common task is certainly semi-supervised node classification in which unlabeled nodes of a given subset are to be classified using a distinct subset of labeled nodes, the train set, from the same graph (Kipf & Welling, 2017; Defferrard et al., 2016). Inductive classification on the other hand refers to the most common setting in machine learning in which nodes to be labeled are not known ahead of time (Hamilton et al., 2017).

A number of architectures have been proposed over the years which deal with specific issues occurring with GNNs. Some combat over-smoothing (which is the tendency for deep GNNs to predict the same labels for all nodes) (Klicpera et al., 2018), some deal with assortativity or heterophily (which refers to situations in which neighboring nodes are likely to have different labels) (Zhu et al., 2020; 2021; Bo et al., 2021) and others still try to learn the connection weights from data using an appropriate attention mechanism (Veličković et al., 2018).

Despite their diversity, these models all have one important shortcoming. Namely they assume that labels can be predicted independently for each node in the graph. In other words they neglect label dependencies altogether. More recently Graph Markov Neural Networks (GMNN) (Qu et al., 2019) were introduced as genuine probabilistic models which include label correlations in graphs by combining the strength of GNNs and those of conditional random fields (CRF) while avoiding their limitations. GMNNs are the models we shall focus on in this work.

The accuracy of the GMNN model was evaluated for node classification and link prediction tasks in Qu et al. (2019) on the classical benchmark datasets Cora, Pubmed and Citeseer (Sen et al., 2008) using the

public splits defined in Yang et al. (2016). Under these settings a clear improvement was demonstrated when comparing the GMNN model to existing baselines that do not account for label dependencies. However, as a number of recent works (Shchur et al., 2018; Errica et al., 2020) have pointed out, a fair evaluation of the performance of GNNs requires a procedure which performs a systematic randomization over train-validation-test set partitions and makes clear separation between model selection and model assessment.

Our aim in this paper is to subject GMNN to such a rigorous performance analysis on a new, relatively large graph of documents with topical labels named *WikiVitals* that we created for that purpose. In a first step we shall rigorously evaluate the effect on the accuracy of a classifier of leveraging the graph structure. This is done by comparing a basic GNN with a graph agnostic baseline such as a MLP. In a second step we shall estimate the increase in accuracy that results from taking into account label correlations using a GMNN on top of a basic GNN. For completeness we also perform the same thorough analysis on the classical benchmark datasets Cora, Citeseer and Pubmed.

In summary, our contributions[1] are:

- We introduce a new dataset of interconnected documents named *WikiVitals* extracted from the English Wikipedia. Compared to the classical benchmark datasets this is a relatively large graph comprising 48k nodes classified into 32 categories and connected by 2.3M edges.

- We apply the fair comparison procedure proposed in Errica et al. (2020) to a GMNN which is a sophisticated node classification model. So far only graph classification models had been evaluated in this manner.

- We evaluate the respective contributions to the accuracy of classifying *WikiVitals* articles when first including the graph structure information using a common GNN and next when leveraging the label correlations information using a GMNN model on top of that GNN.

## 2 Related Work

### 2.1 Modelling Label Dependency in GNNs

Prior to the recent advent of GNNs a number of works had attempted to include label dependencies using various heuristics. Label propagation is such an early attempt where a cost function balances the penalty for predicting the wrong labels with the requirement that node labels should vary smoothly (Zhou et al., 2003; Zhu, 2005).

Dataset specific methods have also been proposed. As far as classifying Wikipedia articles is concerned, authors in Viard et al. (2020) use a simple GNN whose weights are empirically adjusted depending on the similarity of the labels of neighboring nodes. Although these approaches had some empirical success (Huang et al., 2021), the lack of a sound probabilistic foundation makes it difficult to analyze why they fail or succeed. In particular they do not clearly distinguish the contributions of the node features, the graph structure and the label correlations to the prediction accuracy. In our work we decided to avoid using topological node features like node degree, betweenness or assortativity (Newman, 2003; Blondel et al., 2008; Newman, 2005) to make this distinction clearer.

GNNs are a good fit for finding distributed node representations that merge the information supplied by the node features, like the content of a document for instance, with the local structure of the graph in the vicinity of that node. Each such representation is then used for predicting the label for that node independently of those of the other nodes. CRFs on the other hand come in handy for prescribing scores for arbitrary combinations of labels. However performing exact inference is hard due to the trouble of computing the partition function. GMNN propose an elegant solution to this conundrum by using two ordinary GNNs which are coupled when trained with the Expectation Maximization (EM) algorithm (see section 3.1). Their performance was evaluated in Qu et al. (2019) in the usual way using public partitions of classical benchmarks

---

[1]Code and data are available at `https://drive.google.com/file/d/12G_Jx-Wpuko-2KgVfVfS6ZiLK8idW31V/view?usp=sharing`

(Sen et al., 2008) but without accounting for the robustness of this evaluation when using different splits which is essential for a fair evaluation (Errica et al., 2020; Shchur et al., 2018). This is one of our goals. We also evaluate the contributions to the classification accuracy of the three sources of information mentioned above, namely the content of the articles, their interconnections and the correlations between the labels of connected articles.

## 2.2 Evaluating Performance of GNNs

As the authors of Shchur et al. (2018) point out, the evaluations of GNN models are almost never conducted in a rigorous manner. On the one hand, many experiments are not replicable due to the lack of a precise definition of the evaluation process. On the other hand, they argue that using a single split, usually the one defined in the paper that introduces a new benchmark dataset, is insufficient to guarantee the existence of a significant difference between the accuracy of two competing GNN architectures. The authors thus suggest standardizing the choice of hyperparameters and randomizing over many train-validation-test splits. Then they search for a set of hyperparameters that optimizes the average performance over those splits. Surprisingly, they find that the simplest architectures like GCNs (Kipf & Welling, 2017; Defferrard et al., 2016) often perform better for the semi-supervised node classification task than the more sophisticated models (Veličković et al., 2018; Monti et al., 2017).

In our work we follow a still more rigorous accuracy assessment that was originally proposed in Errica et al. (2020) as a SOTA evaluation procedure for the graph classification task. For a given model we search for the best hyperparameters on a per split basis and then average the accuracy estimations of those optimized models over splits. This allows for a fair assessment when comparing two models in the sense that it guarantees that a practitioner who randomly chooses a split, trains her model on the train set, optimizes its hyperparameters on the validation set and estimates the accuracy on the test set will obtain an estimation that is truly reliable for comparing models such as an MLP, a GCN or a GMNN.

## 2.3 Classifying Wikipedia Articles

Wikipedia articles provide rich textual content from which many informative $n$-grams can be extracted in order to build vector representations of the articles. The mutual hyperlinks define a natural graph structure where articles are the nodes of the graph. In this way, several datasets have been created from Wikipedia and are being used to evaluate various GNN architectures, including Squirrel and Chameleon (Rozemberczki et al., 2021)[2]. This is also the case for the *WikiVitals* dataset we introduce in this article.

The labeling of Wikipedia articles can leverage various sources of information. The labels of Squirrel and Chameleon for instance are based on monthly traffic data (acquired through the metadata of the articles) and correspond to an artificial segmentation into 3 or 5 categories (Bo et al., 2021). In Viard et al. (2020), the labeling is based on a collection of labels external to Wikipedia. None of these datasets however exploit a thematic classification resulting from a consensus among Wikipedia editors as does the list of vital articles of Wikipedia[3]. This is the data we used to label the nodes of our *WikiVitals* dataset (see section 4.2). This classification of vital articles, where each document is associated with a unique label, is not exempt of arbitrariness however. Indeed, the assignment of an article to one category or another can sometimes be ambiguous. Furthermore, this classification is imbalanced and contains categories with very few representatives.

A common feature of Wikipedia datasets (Squirrel, Chameleon as well as *WikiVitals*) is that they are more disassortative (Newman, 2003) than classical graph datasets[4]. This makes them particularly interesting as benchmarks for a node classification task, as basic models like GCNs show their limits in such disassortative contexts (Bo et al., 2021). Some recent models like $H_2GCN$ or FAGCN have been proposed to overcome this problem and show better performance in those contexts (Bo et al., 2021; Zhu et al., 2020).

---

[2]http://snap.stanford.edu/data/wikipedia-article-networks.html
[3]https://en.wikipedia.org/wiki/Wikipedia:Vital_articles/Level/5
[4]The notion of heterophily is also commonly used and generally means that a small proportion of edges connects nodes sharing the same label.

# 3 Adapting the Fair Comparison Method to GMNN

## 3.1 Training a GMNN

Before delving into the specifics of our evaluation process let us recall the definition of GMNN model. We use the same notations as in Qu et al. (2019) and refer to this work for a thorough justification of the training procedure we sketch below. We consider a graph $G = (V, E, \mathbf{x}_V)$ where $V$ denotes the set of nodes, $E$ the set of edges and $\mathbf{x}_V := \{\mathbf{x}_n\}_{n \in V}$ the set of features associated to each node $n$. We assume that we are given the one-hot encoded labels (for $K$ categories) $\mathbf{y}_L := \{\mathbf{y}_n\}_{n \in L}$ for the nodes in a subset $L \subset V$ and the features $\mathbf{x}_V$ of all nodes. The task we consider is the prediction of the labels $\mathbf{y}_U$ of the remaining unlabeled nodes in $U = V \setminus L$. The GMNN model does two things. First, it specifies a model for the joint probability $p_\phi(\mathbf{y}_L, \mathbf{y}_U | \mathbf{x}_V)$ compatible with a CRF describing correlations between neighboring nodes. Second, it describes a practical training procedure, based on the EM algorithm, for finding the parameters $\phi$ which maximize a variational lower bound on the marginal likelihood $p_\phi(\mathbf{y}_L | \mathbf{x}_V)$ over the observed labels which we quickly summarize.

Training a GMNN requires defining two ordinary GNNs. The first one, denoted by GNN$_\phi$, where $\phi$ is the set of its parameters, describes the conditional distribution $p_\phi(\mathbf{y}_n | \mathbf{y}_{\mathrm{NB}(n)}, \mathbf{x}_V)$ over individual node labels $\mathbf{y}_n$ given the labels of the neighboring nodes denoted by $\mathrm{NB}(n)$ and the node features $\mathbf{x}_V$. It is specified in the usual manner by a softmax applied on a $d$-dimensional node embedding $\mathbf{h}_{\phi,n}$, read off from the last layer of GNN$_\phi$, multiplied by a $K \times d$ learnable matrix $W_\phi$:

$$p_\phi(\mathbf{y}_n | \mathbf{y}_{\mathrm{NB}(n)}, \mathbf{x}_V) = \mathrm{Cat}(\mathbf{y}_n | \mathrm{softmax}(W_\phi \mathbf{h}_{\phi,n})). \tag{1}$$

A second GNN, that we denote by GNN$_\theta$, defines a mean-field variational distribution meant to approximate the posterior $p_\phi(\mathbf{y}_U | \mathbf{y}_L, \mathrm{x}_V)$ in the EM algorithm. It is defined nodewise in a similar way:

$$q_\theta(\mathbf{y}_n | \mathbf{x}_V) = \mathrm{Cat}(\mathbf{y}_n | \mathrm{softmax}(W_\theta \mathbf{h}_{\theta,n})). \tag{2}$$

Intuitively GNN$_\theta$ is a model that completely neglects correlations among labels. These predictions are then adjusted by GNN$_\phi$ which accounts for the correlations between the labels of neighboring nodes, these in turn will correct GNN$_\theta$ within an EM cyclic training procedure. The training process uses the following two objective functions. One is for updating $\theta$ while holding $\phi$ fixed:

$$O_\theta = \sum_{n \in U} \mathbb{E}_{p_\phi(\mathbf{y}_n | \hat{\mathbf{y}}_{\mathrm{NB}(n)}, \mathbf{x}_V)}[\log q_\theta(\mathbf{y}_n | \mathbf{x}_V)] + \sum_{n \in L} \log q_\theta(\mathbf{y}_n | \mathbf{x}_V), \tag{3}$$

where $\hat{\mathbf{y}}_n$ denotes the ground truth label $\mathbf{y}_n$ if $n \in L$ and is sampled from $q_\theta(\mathbf{y}_n | \mathbf{x}_V)$ if $n \in U$. Using the same notations, the other objective function used for optimizing $\phi$ while holding $\theta$ fixed is:

$$O_\phi = \sum_{n \in V} \log p_\phi(\hat{\mathbf{y}}_n | \hat{\mathbf{y}}_{\mathrm{NB}(n)}, \mathbf{x}_V). \tag{4}$$

The first step of training GMNN is to initialize $q_\theta$ by maximizing the last term in (3) for $\theta$. This corresponds to an ordinary GNN trained without accounting for label correlations. The accuracy of this initial $q_\theta$ model will thus provide a baseline to compare with the full GMNN model. Second, fix $\theta$ and maximize $\phi$ in (4), this is the $M$-step. At last, optimize (3) for $\theta$ while holding $\phi$ fixed, this is the $E$-step. Repeat the $M$ and $E$ step until convergence. Experience shows that $q_\theta$ is consistently a better predictor than $p_\phi$ (Qu et al., 2019).

## 3.2 Fair Comparison of GNNs

Recall that our main goal is to rigorously ascertain under which circumstances a GMNN model architecture, which was designed to leverage label correlations, has a higher accuracy when used for classifying articles from the *WikiVitals* dataset than a correlation agnostic model like a GCN or a FAGCN (Bo et al., 2021). We also wish to compare the GCN or FAGCN models with a structure agnostic baseline like an MLP for this same dataset.

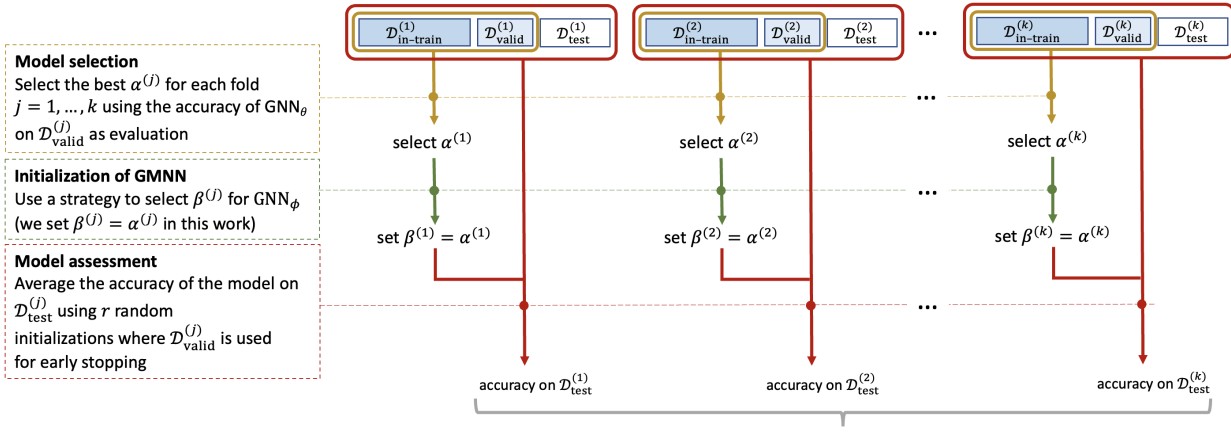

Figure 1: The fair evaluation procedure for GNN's and its adaptation for GMNN uses $k$ train/validation/test splits $\mathcal{D}^{(i)}_{\text{in-train}}, \mathcal{D}^{(i)}_{\text{valid}}, \mathcal{D}^{(i)}_{\text{test}}$ which are created from $k$ stratified folds $\mathcal{F}_i$ as explained in section 3.2.

A crude evaluation would proceed by partitioning the available dataset of labeled *WikiVitals* articles $\mathcal{D} = ((\mathbf{x}_1, \mathbf{y}_1), \ldots, (\mathbf{x}_N, \mathbf{y}_N))$ into three disjoint sets: a train set $\mathcal{D}_{\text{train}}$, a validation set $\mathcal{D}_{\text{valid}}$ for selecting the optimal hyperparameters $\gamma^*$ among a set $\Gamma$ and a test set $\mathcal{D}_{\text{test}}$ to evaluate the accuracy of that optimal model. Unfortunately such a simple procedure was shown to be so unstable that changing the partition could totally scramble relative ranking of various GNN architectures (Errica et al., 2020; Shchur et al., 2018).

To perform reliable comparisons we shall follow the best practices described in Errica et al. (2020). The main requirement is to clearly separate model assessment from model selection.

**Model assessment** uses a $k$-fold cross validation procedure. The dataset $\mathcal{D}$ is first split into $k$ disjoint stratified[5] folds $\mathcal{F}_1, \ldots, \mathcal{F}_k$. Then $k$ different train and test sets are defined as:

$$\mathcal{D}^{(i)}_{\text{train}} := \bigcup_{j \neq i} \mathcal{F}_j, \quad \mathcal{D}^{(i)}_{\text{test}} := \mathcal{F}_i, \quad i = 1, \ldots, k.$$

Each train set is itself split into an inner train set and a validation set:

$$\mathcal{D}^{(i)}_{\text{train}} := \mathcal{D}^{(i)}_{\text{in-train}} \cup \mathcal{D}^{(i)}_{\text{valid}}, \quad i = 1, \ldots, k.$$

Model selection (see below) is performed separately for each $\mathcal{D}^{(i)}_{\text{train}}$. This results in a set of hyperparameters $\gamma^{(i)}$ which is optimal for $\mathcal{D}^{(i)}_{\text{train}}$. The model is then trained with these optimal hyperparameters $\gamma^{(i)}$ on $\mathcal{D}^{(i)}_{\text{in-train}}$ using $\mathcal{D}^{(i)}_{\text{valid}}$ to implement early stopping. Actually, for each fold $i$, the test accuracy is averaged over $r$ training runs with different random initializations of the weights to smooth out any possible bad configuration. The average of these test accuracies over the $k$ folds makes our final assessment of a model architecture.

**Model selection** corresponds to choosing an optimal set of hyperparameters. It is performed, separately for each $\mathcal{D}^{(i)}_{\text{train}}$. More precisely, the model is trained on $\mathcal{D}^{(i)}_{\text{in-train}}$ using $\mathcal{D}^{(i)}_{\text{valid}}$ as a holdout set for selecting the hyperparameters $\gamma^{(i)}$ which maximize the accuracy among a set $\Gamma$ of configurations.

### 3.3 Adaptation to GMNN

In order to evaluate GMNN using the fair evaluation principle described above, we must select for each split $j$ a pair $\gamma^{(j)} := (\alpha^{(j)}, \beta^{(j)})$ of optimal hyperparameters $\alpha^{(j)}$ for $\text{GNN}_\theta$ and $\beta^{(j)}$ for $\text{GNN}_\phi$ respectively. We

---

[5]In the context of this paper, stratified sampling of nodes means that the distribution of labels in the sampled set is approximately the same as in the set of nodes from which it was sampled.

follow the same strategy as Qu et al. (2019) namely we compute hyperparameters $\alpha^{(j)}$ for $\text{GNN}_\theta$ first and then set $\beta^{(j)} = \alpha^{(j)}$ for the hyperparameters of $\text{GNN}_\phi$. The model assessment phase remains unchanged. For each split $j$ the set of hyperparameters $\gamma^{(j)} = (\alpha^{(j)}, \beta^{(j)})$ is used to compute the test accuracy of GMNN using $r$ random weight initializations. The fair evaluation model adapted to GMNN is presented in figure 1.

Note that performing a fair evaluation of the model after completion of the initial training of $\text{GNN}_\theta$ and before entering the EM optimization corresponds to a fair evaluation of a plain GNN which thus requires no additional computation.

## 4 Experiment

### 4.1 Datasets and Settings

**Datasets:** For our main experiment we created *WikiVitals*, a novel sparse and disassortative document-document graph created from the English Wikipedia level 5 vital articles in April 2022 (Wikipedia is under CC BY-SA license). Node features $\mathbf{x}_n$ correspond to the presence or absence of 4000 $n$-grams in the summary section, title and headings of the article. The edges are the hyperlinks between articles. Each node of the graph has been associated to a single label (among 32) corresponding to an intermediate level in a hierarchy of topics co-constructed by Wikipedia editors. More information on this new dataset as well as statistics on all datasets can be found in table 1 and section 4.2. For completeness, we also performed a fair evaluation on the three well-known assortative citation network datasets: Cora, Citeseer, and Pubmed. Edges in these networks represent citations between two scientific articles, node features $\mathbf{x}_n$ are a bag-of-words vector of the articles and labels $\mathbf{y}_n$ correspond to the fields of the articles. For all datasets, we treat the graphs as undirected.

**General setup:** All baseline models (MLP, GCN and FAGCN) were reimplemented using PyTorch with two layers (input representations $\rightarrow$ hidden layer $\rightarrow$ output layer). For all models, $L^2$-regularization is performed on all layers, dropout is applied on input data and on all layers. For GCN and FAGCN, we used the so-called *renormalization trick* of the adjacency matrix (Kipf & Welling, 2017). For FAGCN, the number of propagations (Bo et al., 2021) is set to 2 in order to limit the aggregation of information to nodes located at a maximum distance of 2. For GMNN we use the annealing sampling method with factor set to 0.1 (Qu et al., 2019), the number of EM-loops is set to 10. Both label predictions $\hat{\mathbf{y}}_n$, made with $\text{GNN}_\theta$, and node features $\mathbf{x}_V$ are used to train $\text{GNN}_\phi$ as defined in (4).

We use the same training procedure for all models. For all datasets, node features are binarized and then normalized ($L^1$-norm) before training. We used the Adam optimizer (Kingma & Ba, 2015) with default parameters and no learning rate decay, the same maximum number of training epochs, an early stopping criterion and a patience hyperparameter (see section 4.3 for more details). Validation accuracy is evaluated at the end of each epoch. All model parameters (convolutional kernel coefficients for FAGCN, weight matrices for all models) are initialized and optimized simultaneously (weights are initialized according to Glorot and biases initialized to zero). In all cases we use full-batch training (using all nodes in the training set every epoch).

**Fair evaluation setup:** During the assessment phase, we perfom $r = 20$ trainings for each of the $k = 10$ splits. Best configurations of hyperparameters $\alpha^{(j)}$ for $\text{GNN}_\theta$ are calculated for each split $j$, and next we set hyperparameters $\beta^{(j)} = \alpha^{(j)}$ for $\text{GNN}_\phi$.

For each dataset, we followed the best practices advocated in Errica et al. (2020) and summarized in section 3.2 to pre-calculate stratified splits $(\mathcal{D}^{(j)}_{\text{in-train}}, \mathcal{D}^{(j)}_{\text{valid}}, \mathcal{D}^{(j)}_{\text{test}})$, $j = 1, \ldots, k$ of the entire set of nodes with respective ratios of 81%, 9% and 10%. In the sequel the sets $\mathcal{D}^{(j)}_{\text{in-train}}$ will be referred to as dense training sets.

In addition, we have created two other sets of splits whose train sets are sparse, first to allow a convenient comparison with previous work which actually use such train sets (Yang et al., 2016), second to enlarge the scope of the methods tested in this article. As a reminder, the evaluation of GNNs as well as GMNN

| Dataset | Assortativity | #Nodes | #Edges | #Categories | #Features |
|---|---|---|---|---|---|
| Cora | 0.771 | 2,708 | 5,429 | 7 | 1,433 |
| Citeseer | 0.675 | 3,327 | 4,732 | 6 | 3,703 |
| Pubmed | 0.686 | 19,717 | 44,338 | 3 | 500 |
| WikiVitals | 0.204 | 48,512 | 2,297,782 | 32 | 4,000 |

Table 1: Statistics of document graphs

for Cora, Citeseer and Pubmed was classically performed using the Planetoid splits (Yang et al., 2016) of these datasets or similarly constructed splits composed of 20 nodes per category randomly selected in the whole dataset (Shchur et al., 2018; Bo et al., 2021; Qu et al., 2019). To construct splits with sparse train sets we independently extracted two subsets $\mathcal{D}^{(j)}_{\text{sparse-balanced}}$ and $\mathcal{D}^{(j)}_{\text{sparse-stratified}}$ from each $\mathcal{D}^{(j)}_{\text{in-train}}$, $j = 1, \ldots, k$. Each contains $20 * K$ nodes (where $K$ is the number of categories). Each $\mathcal{D}^{(j)}_{\text{sparse-balanced}}$ is constructed by selecting 20 nodes of each category from $\mathcal{D}^{(j)}_{\text{in-train}}$. In the sequel these sets will be referred to as sparse balanced train sets in the sense that each category is represented equally in each of them. Each $\mathcal{D}^{(j)}_{\text{sparse-stratified}}$ is constructed by selecting nodes from $\mathcal{D}^{(j)}_{\text{in-train}}$ in a stratified way. We shall denote these sets as sparse stratified train sets. Thus we have $k$ splits of each dataset with sparse balanced train sets $(\mathcal{D}^{(j)}_{\text{sparse-balanced}}, \mathcal{D}^{(j)}_{\text{valid}}, \mathcal{D}^{(j)}_{\text{test}})$, $j = 1, \ldots, k$ and $k$ splits of each dataset with sparse stratified train sets $(\mathcal{D}^{(j)}_{\text{sparse-stratified}}, \mathcal{D}^{(j)}_{\text{valid}}, \mathcal{D}^{(j)}_{\text{test}})$, $j = 1, \ldots, k$. The fair evaluation method presented in section 3.3 can be easily adapted to splits with sparse train sets replacing the inner-train sets in every training phases.

Rigorous model selection phases imply performing extensive grid searches over the hyperparameter set $\Gamma$, which is computationally very expensive. In practice we have implemented our own evolutionary grid search algorithm which discovers suitable configurations of hyperparameters by using the validation accuracy to guide the evolution. Such an algorithm computes a suitable configuration by exploring a small portion of $\Gamma$ (Young et al., 2015), see section 4.3 for more details.

## 4.2 Creation of the WikiVitals dataset

*WikiVitals* is a disassortative document-document network created from 48512 vital Wikipedia articles extracted from a complete Wikipedia dump dated April 2022. Nodes correspond to vital Wikipedia articles. Node features are binary bag-of-words sparse representations of the articles. Each of the 4000 features in these representations corresponds to the presence or absence of an informative unigram or bigram in the introduction, title or section titles of the article. Edges correspond to the mutual hyperlinks between articles in the corpus of vital articles.

Vital articles have been selected by Wikipedia editors and have been categorized by topic. We extracted a 3-level hierarchy of topics and used the 32 intermediate topics within this hierarchy as labels assigned to each node in the graph. Each node was assigned a single label. The table 2 shows a partial view of the topic hierarchy, focusing on the 32 intermediate categories we used in this paper[6].

We relied on a dump of the English Wikipedia because it provides a frozen view of the English Wikipedia and because it contains the text of the articles. Using the Wikidata knowledge graph instead would have lead to a similar dataset but with a different processing.

We conducted feature extraction on the abstracts, titles and headers for the vital articles. To preprocess this data, we ignored the stop words using nltk.corpus and applied stemming using Snowballstemmer from nltk. Next, we extracted the most frequent unigrams and bigrams from the abstracts and headers with a frequency greater than $1 \times 10^{-3}$ and from the titles with a frequency greater than $1 \times 10^{-4}$. Finally, we retained the top 4000 features that were most predictive of the labels in the chi-squared sense.

---

[6]The top level of the hierarchy comprises 11 coarse topics, the middle level 32 topics and the finest level 251 topics.

| Class name | #articles |
|---|---|
| *Arts* | |
| 01-Arts | 3310 |
| *Biological and health sciences* | |
| 02-Animals | 2396 |
| 03-Biology | 886 |
| 04-Health | 791 |
| 05-Plants | 608 |
| *Everyday life* | |
| 06-Everyday life | 1191 |
| 07-Sports, games and recreation | 1231 |
| *Geography* | |
| 08-Cities | 2030 |
| 09-Countries | 1386 |
| 10-Physical | 1902 |
| *History* | |
| 11-History | 2979 |
| *Mathematics* | |
| 12-Mathematics | 1126 |
| *People* | |
| 13-Artists, musicians, and composers | 2310 |
| 14-Entertainers, directors, producers, and screenwriters | 2342 |
| 15-Military personnel, revolutionaries, and activists | 1012 |
| 16-Miscellaneous | 1186 |
| 17-Philosophers, historians, political and social scientists | 1335 |
| 18-Politicians and leaders | 2452 |
| 19-Religious figures | 500 |
| 20-Scientists, inventors, and mathematicians | 1108 |
| 21-Sports figures | 1210 |
| 22-Writers and journalists | 2120 |
| *Philosophy and religion* | |
| 23-Philosophy and religion | 1408 |
| *Physical sciences* | |
| 24-Astronomy | 886 |
| 25-Basics and measurement | 360 |
| 26-Chemistry | 1207 |
| 27-Earth science | 849 |
| 28-Physics | 988 |
| *Society and social sciences* | |
| 29-Culture | 2075 |
| 30-Politic and economic | 1825 |
| 31-Social studies | 355 |
| *Technology* | |
| 32-Technology | 3148 |

Table 2: The 32 labels of the nodes of the *WikiVitals* dataset classified by topics of higher granularity

### 4.3 Hyperparameters, training, and grid search

**Hyperparameters and search space**: Grid search during model selection was performed over the following search space $\Gamma$:

- hidden dimension: $[8, 16, 32, 64]$
- input dropout: $[0.2, 0.4, 0.6, 0.8]$
- dropout: $[0.2, 0.4, 0.6, 0.8]$
- learning rate: [1e-1, 5e-2, 1e-2, 5e-3, 1e-3, 5e-4, 1e-4]
- $L^2$-regularization strength: [1e-1, 5e-2, 1e-2, 5e-3, 1e-3, 5e-4, 1e-4, 5e-5, 1e-5]
- $\epsilon$ (only for FAGCN): $[0.1, 0.2, 0.3, 0.4, 0.5, 0.6, 0.7, 0.8, 0.9]$

For *WikiVitals*, we use a reduced search space, hidden dimension was set to 64 and $L^2$-regularization strength to 1e-5, learning rate was in [1e-1, 5e-2], and $\epsilon$ in $[0.7, 0.8, 0.9]$.

**Training procedures for GNN models**: For all GNN model training:

- We train for a maximum of 1000 epochs.
- We use early stopping, patience is set to 200.
- There is no learning rate decay.
- An $L^2$-regularization is applied on all layers.
- All model parameters (convolutional kernel coefficients for FAGCN, weight matrices for all models) are optimized simultaneously.
- Once training has stopped, we reset the state of model parameters to the step with the lowest validation loss.

For MLP and GCN, we use early stopping which stops the optimization if the validation loss does not decrease for 200 epochs. For FAGCN, we to stop the optimization if the validation loss and the validation accuracy do not decrease for 200 epoches. For GMNN training, we train models for 100 epoches and perform 10 EM-loops.

**Evolutionary grid search**: Our evolutionary algorithm maintains a randomly initialized population of 100 configurations of hyperparameters over generations. At each generation we retain between 2 and 50 configurations whose validation accuracy exceeds the population average for the next generation. New configurations are generated via a 2-pivot crossover, two configurations that have a better evaluation being more likely to be selected. A mutation step assigns a new value to a configuration hyperparameter with a probability 0.05 to promote exploration. Only configurations never seen before are added to complete the population at each generation. The number of generations is set to 10. Beyond this value the evaluation of the best configurations in the population does not seem to increase significantly.

## 5 Results

Quantitative results for the node classification task applied to our *WikiVitals* dataset and to the classical Cora, Citeseer and Pubmed datasets are presented in tables 3 and 4. To account for the unfortunate possibility that the EM phases could perhaps decrease the accuracy after the initialization phase we retain the best accuracy among the EM phases only. We thus compare the average accuracies over the $k$ splits before and after the EM phases. More precisely, we perform a relational $t$-test between those paired means where the alternate hypothesis is that the accuracy after the EM phase is higher than before. Notation for significance in tables 3 and 4 using $p$-value are: *** if $p < 0.001$, ** if $p < 0.01$, * if $p < 0.05$.

**Leveraging the graph structure with GNNs**: These results confirm that taking into account the underlying graph structure provides a significant performance gain for the node classification task for all datasets, regardless of whether the train set is dense or sparse. For classical datasets, the GCN and FAGCN models outperform the use of an MLP which only takes into account node features disregarding the graph structure. For *WikiVitals*, which is a disassortative dataset, the FAGCN model performs best. Actually in this case, when using dense train sets for training, a simple MLP performs better than the basic GCN (see table 4). We thus confirm the importance to choose an architecture adapted to the level of assortativity of the graph.

**Leveraging label correlations with GMNN**: Referring to tables 3 and 4, a general observation is that using GMNN for *WikiVitals*, Cora, Citeseer, Pubmed leads to the best average performance, whether the train sets are sparse or dense.

For dense train sets the improvement provided by GMNN is either small, but nevertheless significant, or it is insignificant. Practically, when a large proportion of the dataset is available for training a model GMNN could be worth a try. Yet this small improvement should be balanced against the high computation cost incurred.

| | Cora | | Citeseer | | Pubmed | | Wikivitals | |
|---|---|---|---|---|---|---|---|---|
| | balanced train set | stratified train set | balanced train set | stratified train set | balanced train set | stratified train set | balanced train set | stratified train set |
| MLP | 58.54 (3.98) | 58.32 (2.17) | 59.84 (3.54) | 58.35 (2.48) | 71.23 (2.85) | 70.39 (1.70) | 68.60 (0.92) | 69.35 (1.10) |
| GNN (base) | 80.78 (2.58) | 81.31 (2.16) | 69.05 (3.66) | 70.94 (2.16) | 80.20 (1.88) | 80.50 (2.38) | 70.64 (0.85) | 72.68 (1.17) |
| + GMNN $p_\phi$ | 81.14 (3.19) | 81.56 (2.38) | 67.04 (7.47) | 70.82 (2.54) | 81.26 (1.34) | 81.15 (2.30) | 74.72 (1.19) | 74.64 (1.38) |
| $q_\theta$ | 80.76 (3.74) | 81.56 (2.30) | 69.34 (3.96) | 71.52 (2.19) | 81.55 (1.43) | 81.60 (2.53) | 74.77 (1.18) | 74.69 (1.37) |
| best | **81.67 (3.00)** | **81.91 (2.21)** | **69.61 (3.96)** | **71.62 (2.20)** | **81.67 (1.32)** | **81.70 (2.45)** | **74.80 (1.18)** | **74.73 (1.36)** |
| Significance | ** | * | * | * | ** | *** | *** | *** |

Table 3: Test accuracy reported in % using sparse train sets. Best results are highlighted. The base GNN is GCN for Cora, Citeseer and Pubmed, it is FAGCN for *WikiVitals*.

| | Cora | Citeseer | Pubmed | WikiVitals |
|---|---|---|---|---|
| MLP | 78.49 (2.39) | 75.02 (2.15) | 88.68 (0.86) | 86.55 (0.42) |
| GCN | 88.84 (2.39) | 77.24 (1.73) | 89.20 (0.86) | 72.74 (0.61) |
| + GMNN | **89.26 (1.91)** | 77.43 (1.70) | 89.18 (0.84) | 74.19 (0.42) |
| Significance | * | | | *** |
| FAGCN | 88.87 (1.99) | 78.27 (3.53) | 90.23 (0.90) | 87.84 (0.32) |
| + GMNN | 89.08 (1.76) | **78.32 (3.64)** | **90.34 (0.88)** | **87.92 (0.31)** |
| Significance | * | | *** | *** |

Table 4: Test accuracy reported in % using dense train sets. Best results are highlighted.

For sparse train sets GMNN brings a more obvious improvement to the accuracy. This is true for all the datasets that were analyzed. More precisely, this improvement is significant when comparing GMNN with GCN on classical datasets and when comparing it with FAGCN on *WikiVitals*.

Considering Table 3 we notice that the GMNN accuracies for either the balanced or the stratified sparse train sets of Cora, Pubmed and *WikiVitals* are almost equal. Thus the EM iterations in GMNN seem to converge to the optimal accuracy provided that the model already performed well enough on the baseline. The poor improvement observed on Citeseer for balanced train sets will be discussed below in this section.

The very significant improvement brought by GMNN for *WikiVitals* when using a sparse train set may be interpreted as follows. In such a situation the information supplied by the correlations between labels seems particularly useful to compensate for the small number of nodes available for training. On the other hand, when the train set is dense, the information for making accurate predictions is supplied by the features of a large number of nodes.

**The cost of a fair evaluation:** The fair evaluation method is a computationally expensive method, especially in the model selection phase. Model selection and model assessment for Wikivitals, which is the largest dataset we considered, required roughly 30 GPU hours. We limit this cost using three different techniques. First, when training the model on *WikiVitals* we cannot afford exploring the whole hyperparameters space $\Gamma$ that was defined for Cora, Citeseer and Pubmed. We thus restrict the exploration to a subset instead, as described in 4.3. A second approach is to to use an evolutionary grid search algorithm which limits the exploration to roughly $2\% - 15\%$ of $\Gamma$ depending on the model that was described in 4.3. At last, we can chose an parsimonious strategy for selecting the hyperparameters $\beta^{(j)}$ once $\alpha^{(j)}$ has been determined, which we did by setting $\beta^{(j)} = \alpha^{(j)}$.

**The limits of a fair evaluation:** A possible issue with the fair evaluation could occur when the validation sets $\mathcal{D}_{\text{valid}}^{(j)}$ are too small. In such situations the selected hyperparameters $\gamma^{(j)}$ are at higher risk to be sub-optimal. To address this we could obviously adopt the same strategy that was used in the evaluation phase, namely randomizing over several initializations of the model parameters. We think this is the origin of the poor performance observed on Citeseer with the balanced train set in table 3.

# 6 Conclusion and perspectives

This paper introduces a new large and disassortative document-document graph dataset named *WikiVitals* and adapts a fair comparison method of GNNs to GMNN to evaluate the contribution of three distinct sources of information for a semi-supervised node classification task: the node features, the underlying graph structure and the label correlations. Taking into account label correlations via a GMNN model was our main focus. We confirmed that these correlations indeed provide a significant improvement for the classification accuracy when the nodes available during training are sparse over the entire graph. The results were observed for both *WikiVitals* and classical benchmark datasets Cora, Citeseer and Pubmed. This makes us confident that this conclusion holds for a practical use of GMNN.

In future work we intend to leverage the hierarchical categorization that comes with the *WikiVitals* dataset to improve classification accuracy. The question of how exactly to properly define correlations for such hierarchical labels is much more subtle. Another interesting research direction would be handling the multi-label case. This requires both defining an appropriate notion of multi-label correlation and adapting the GMNN model accordingly. This would for instance allow us to deal with ambiguous situations in which some documents are assigned several labels, which often occurs in practice.

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
