# OpenReview forum: "Fair Evaluation of Graph Markov Neural Networks"
_TMLR — Rejected by TMLR_

### Review · Reviewer_Bp7f · 2023-09-16

**Summary Of Contributions:**

The paper presents a new dataset for node classification, gathered from Wikipedia, in an effort to evaluate Graph Markov Neural Networks , a previously introduced model, in a more principled, clear, and reproducible manner. The authors adjust a fair evaluation methodology that was originally formulated for graph classification and is based on utilizing k-fold cross-validation.   The proposed dataset stands out from the other Wikipedia graph learning datasets because the node labeling is based on editors. The authors also provide an in-depth summary and a justification of their focus on the GMNN model. Overall, the paper argues in favor of an alternative, improved evaluation manner in node classification tasks, formulates it, and tests it on a previously introduced model on a new dataset gathered specifically for the paper in an attempt to make GNN research more rigorous and enhance reproducibility.

**Audience:**

Yes

**Broader Impact Concerns:**

I detected no ethical concern about the broader impact of the paper.

**Claims And Evidence:**

Yes

**Requested Changes:**

1. As mentioned above, to my understanding, the evaluation methodology is the main outcome of the paper. In this case, for an objective evaluation and to testify to the generalization of the method, the authors should 1. add more GNN baselines and 2. clarify the difference between the traditional evaluation and the current evaluation with quantitative or qualitative means in the results section.
2. I would suggest diminishing the GMNN part (e.g. section 3.1 ), because it does not seem directly connected to the arguments of the paper, meaning that the evaluation method can be used in all node classification GNNs. It would be more meaningful to argue further on the usefulness of the evaluation process with more toy examples or more experiments.
3. The WikiVitals dataset is an important part of the paper, and some further information about it i.e. graph statistics, homophily percentage, correlation between features and labels etc. would support its usage.
4. “we retain the best accuracy among the EM phases only” If this is performed during testing, the evaluation may be erroneous. If not, it should be clarified.

**Strengths And Weaknesses:**

Strengths:
1. The paper succeeds in motivating and proposing a more objective method of evaluation that is based on adjusting a previous work.
2. The new Wikipedia dataset is a contribution, as carefully constructed datasets are an indispensable part of advancing research in the field.
3. The code is provided for reproducibility.

Weaknesses:
1. The way the paper is presented, the contribution seems to be the evaluation of GMNN, which is a previously published model. If the contribution is a general new evaluation of GNNs with hyperparameters, along with a new helpful dataset,  it should be clarified and disconnected from the model.
2. The results do not adequately indicate the difference between the old evaluation and the new one but rather argue about the usefulness of GMNN. The new evaluation does not seem to impact the experiment outcome in practice e.g. the ranking of the methods.

---

> ### Author Response · Authors · 2023-09-27
>
> Here are our responses:
>
> 1. Our focus was to use a SOTA evaluation method (Errica et al., 2020) to examine the various contributions of different sources of information (content, links and label correlations) to the accuracy of a node classifier in the setting of a large and realistic graph of documents. Regarding your request to clarify the difference with traditional evaluation: we think it does not make sense to do this in a quantitative way because the method we used performs averages of per-split optimal parameter selection which traditional methods do not. We acknowledge however that this evaluation does not change the ranking of the MLP, GNN and GMNN models.
>
> 2. We are ready to diminish section 3.1 but we consider this would negatively impact the readability. This part is indeed related to the main focus of our paper which, among other things, was intended to evaluate the contribution of the correlations between node labels that these GMNN models take care of properly.
>
> 3. Key information regarding the Wikivitals dataset was provided in Table 1. We chose to report assortativity rather than homophily as both refer the statistics of neighboring labels.
>
> 4. We acknowledge that this sentence introduces confusion and we are considering a rewrite of the paragraph in which it is found. As indicated in the last paragraph of section 3.3, the evaluation of both the basic GNN and GMNN is carried out so as to compare them. Model parameters used for evaluation (whether for GNN or GMNN) were determined, as indicated in section 4.3, during the training phase on the basis of minimum validation loss.

---

> > ### Comment · Reviewer_Bp7f · 2023-10-05
> > **Answer to the authors**
> >
> > Thank you for your answer. However, in order to argue further about the applicability of the evaluation procedure, as I mentioned above there needs to be more convincing proof that it is beneficial to what is currently being done i.e. there is a big enough impact on the results and the conclusions. Moreover, to be considered a solid contribution, its use should be more general i.e. testing it for other models.

---

### Review · Reviewer_v23K · 2023-09-19

**Summary Of Contributions:**

The authors provide an extended evaluation of Graph Markov Neural Networks and provide a new dataset, called WikiVitals.

**Audience:**

No

**Broader Impact Concerns:**

I have no concerns regarding the broader impact.

**Claims And Evidence:**

Yes

**Requested Changes:**

See weaknesses above.

I also recommend to provide the source code online. Particularly in such an experiment focused work, the reassurance whether the code reproduces the results is evident.

Minor:
- In the second paragraph you write, „The most common task is certainly semi-supervised node classification“. I personally doubt this is true. In many domains domains, including quantum chemistry and NLP, this task is actually non-existent.
Please write at least something like „A very common task is arguably semi-supervised node classification“

**Strengths And Weaknesses:**

Strength:
- well written
- well structured

Weaknesses:
The contribution is quite shallow! E.g. the ideas for the fair comparison seems to be fully adapted, and not very different to what has been done on GNNs. I.e. the adaptation seems straight forward. Also the model is not new and the formulas in Sect. 3.1 seem to be just copied from Qu et al. 2019.

---

> ### Author Response · Authors · 2023-09-27
>
> Here are our responses:
>
> - "Weaknesses: The contribution is quite shallow"
>
> We acknowledge that we do not introduce a new model and that we use an already existing evaluation method, but we never claimed otherwise. However, we did introduce a new, large document graph dataset that we believe could be useful in this field. We conducted a rigorous evaluation (which goes beyond cross-validation) that had not been done before and can be useful for practitioners. The formulas in section 3.1 were provided primarily to improve the readability of the paper. We also took care to use the same notations as in the seminal work on GMNN.
>
> - "I also recommend to provide the source code online."
>
> The source code was attached to the submission. In addition, we will of course include a link to github in the article, which we haven't done for anonymity reasons.
>
> - "In the second paragraph you write, „The most common task is certainly semi-supervised node classification“. [...] Please write at least something like „A very common task is arguably semi-supervised node classification“
>
> We will follow up your request for semi-supervised node classification.

---

### Review · Reviewer_7EsT · 2023-09-19

**Summary Of Contributions:**

The authors present a novel dataset, WikiVitals, tailored for transductive semi-supervised node classification. This dataset encompasses a graph containing 48k Wikipedia articles, with each node/article classified into one of 32 distinct categories. In addition, an evaluation of the Graph Markov Neural Networks on the WikiVitals dataset is conducted.

**Audience:**

No

**Claims And Evidence:**

No

**Requested Changes:**

Considering the primary focus of the paper is the unveiling of a new dataset, TMLR may not be the most suitable journal for this submission.

**Strengths And Weaknesses:**

#### Strengths
- Introduction of the WikiVitals dataset, a valuable addition to the field of transductive semi-supervised node classification.

#### Weaknesses
- The central contribution of this paper is the introduction of a new dataset. However, I believe that TMLR may not be the most appropriate journal for this type of contribution. The authors might consider submitting to DMLR, which may be more fitting.
- While a fair evaluation of GNNs is commendable, the paper focuses exclusively on document-document networks and is limited to small or medium-scale datasets. Extending the evaluation to include datasets from [OGB's large-scale datasets](https://ogb.stanford.edu/docs/nodeprop/) would enhance the study's comprehensiveness.
- The field already boasts extensive research on the fair evaluation of GNNs, either for graph classification (Dwivedi et al., 2020) or node classification (Zhao et al., 2020). It is crucial that this study acknowledges, cites, and articulates its distinction from these existing works.

#### Reference
Zhao, Wentao, et al. "A pipeline for fair comparison of graph neural networks in node classification tasks." arXiv preprint:2012.10619 (2020).

Dwivedi, Vijay Prakash, et al. "Benchmarking graph neural networks." arXiv preprint arXiv:2003.00982 (2020), published at JMLR 2022.

---

> ### Author Response · Authors · 2023-09-27
>
> Here are our responses:
>
> - "I believe that TMLR may not be the most appropriate journal for this type of contribution. The authors might consider submitting to DMLR, which may be more fitting."
>
> We chose to submit our work to TMLR because reproducibility studies and accounts of applications of existing techniques are within the scope of this journal. We acknowledge that DMLR could be a fitting alternative.
>
> - "While a fair evaluation of GNNs is commendable, the paper focuses exclusively on document-document networks and is limited to small or medium-scale datasets. Extending the evaluation to include datasets from OGB's large-scale datasets would enhance the study's comprehensiveness."
>
> The Wikivitals dataset we are presenting is considerably larger than the typical benchmark datasets (Cora, Citeseer, Pubmed) used for evaluating GNNs in general. Handling even larger datasets like those you suggested (OGB) is beyond our computing resources, especially because the models being analyzed are not well-suited for parallel processing.
>
> - "The field already boasts extensive research on the fair evaluation of GNNs, either for graph classification (Dwivedi et al., 2020) or node classification (Zhao et al., 2020)."
>
> We acknowledge this and will include these works in the related work section 2.2 and highlight the specificity of our work, which is to evaluate the contribution to accuracy of different sources of information.

---

### Review · Reviewer_wYbF · 2023-09-21

**Summary Of Contributions:**

In this paper the authors release a new dataset named WikiVitals for node classification task. Graph Markov Neural Networks (GMNN) model is applied to the dataset and the authors conduct fair comparison procedure which involves multi-fold cross validation. Ablation study is conducted to evaluate the respective contributions to the classification accuracy from regular GNN and GMNN.

**Audience:**

No

**Broader Impact Concerns:**

I have no concerns regarding the ethical implications.

**Claims And Evidence:**

Yes

**Requested Changes:**

I'd recommend the authors to highlight the novel contributions in this work.

**Strengths And Weaknesses:**

**Strengths**
* The paper release a new dataset which is relatively large and provides sufficient details on data stats and how the data is created.
* The discussion of related works seems extensive.
* Empirical settings is discussed in detail, including hyper parameter selection and NN architecture selection.

**Weaknesses**
* It is not clear to me what the main contribution, other than releasing a new dataset. Neither the neural network architecture (GMNN) nor the evaluation procedures are novel. Overall the paper reads like a technical/empirical report than a research paper.
* The 'fair' model assessment method seems to be plain cross validation procedure to me. Maybe I am missing something but evaluationg a machine learning model with K fold cross validation is a straightforward idea and I fail to see the novelty in it.

---

> ### Author Response · Authors · 2023-09-27
>
> Here are our responses:
>
> - "It is not clear to me what the main contribution, other than releasing a new dataset."
>
> Our contribution beyond the introduction of Wikivitals is to assess in a fair way the contribution to accuracy of various sources of information (document features, graph structure information and correlations between labels). Moreover, we chose to submit our work to TMLR because reproducibility studies and accounts of applications of existing techniques are within the scope of this journal. Our contributions, fit within this scope.
>
> - "The 'fair' model assessment method seems to be plain cross validation procedure to me."
>
> That is not true. The fair assessment procedure advocated in Errica et al., 2020 promotes a separate evaluation for each split, as we explain in detail in the second paragraph of section 2.2 and the third part of section 3.2. This has never been done before, especially for such a large and typical graph of documents, such as Wikivitals

---

### Decision · Action_Editor_7j7M · 2023-10-18

**Recommendation:** Reject

**Comment:**

The reviewers unanimously recommend rejection. The main concerns raised by the reviewers are as follows:

- One of the main claims of the paper is that label correlations provide a significant improvement in performance when the nodes available during training are sparse over the entire graph. However, this claim is not supported by convincing evidence since the different methods are only evaluated on document-document networks and these networks are relatively small.

- The related work section is not complete. A lot of recent work has focused on the fair evaluation of GNNs (see for instance [1], [2] and [3]). A discussion of these methods is missing from the paper.

- There is a lack of explanations in the paper. More specifically, it is not clear how this work is different from standard evaluation protocols and more details need to be provided.

[1] Dwivedi, V. P., Joshi, C. K., Luu, A. T., Laurent, T., Bengio, Y., & Bresson, X. (2023). Benchmarking Graph Neural Networks. Journal of Machine Learning Research, 24(43), 1-48.\
[2] Zhao, W., Zhou, D., Qiu, X., & Jiang, W. (2020). A pipeline for fair comparison of graph neural networks in node classification tasks. arXiv preprint arXiv:2012.10619.\
[3] Dwivedi, V. P., Rampášek, L., Galkin, M., Parviz, A., Wolf, G., Luu, A. T., & Beaini, D. (2022). Long range graph benchmark. Advances in Neural Information Processing Systems, 35, 22326-22340.

**Audience:**

The topic is relevant to the TMLR audience, however, I do not think that a lot of people would be interested in knowing the findings of this paper. This is mainly because the work:
- re-introduces an evaluation framework that has already been published before (Errica et al. (2020))
- evaluates the GMNN model solely on document-document networks

**Claims And Evidence:**

The main claims of contribution in the paper include:
- the introduction of a new dataset of interconnected documents extracted from Wikipedia.
- a framework for the fair comparison of the GMNN node classification model (adapted from Errica et al. (2020)).
- an evaluation of the performance improvement brought by label correlations information.

Overall, I think that more extensive experiments are needed to support the last claim.

**Resubmission Of Major Revision:**

The authors may consider submitting a major revision at a later time.